

# Design of parametric risk transfer solutions for volcanic
# eruptions: an application to Japanese volcanoes
Delioma Oramas-Dorta[1], Giulio Tirabassi[1], Guillermo E. Franco[1], Christina Magill[2]
[1] Guy Carpenter & Co, LLC. Tower Place West, London, EC3 5BU, United Kingdom.
[2] Department of Environmental Sciences, Faculty of Science and Engineering, Macquarie University, NSW 2109,
Australia.
*Correspondence to*: Dr. Delioma Oramas-Dorta (Delioma.Oramas-Dorta@guycarp.com).



**Abstract**
Volcanic eruptions are rare but potentially catastrophic phenomena, affecting societies and economies through
different pathways. The 2010 Eyjafjallajökull eruption in Iceland, a medium-sized ash fall producing eruption,
caused losses in the range of billions of dollars, mainly to the aviation and tourist industries. Financial risk transfer
mechanisms such as insurance are used by individuals, companies, Governments, etc. to protect themselves from
losses associated to natural catastrophes. In this work, we conceptualize and design a parametric risk transfer
mechanism to offset losses to building structures arising from large, ash fall-producing volcanic eruptions. Such
transfer mechanism relies on the objective measurement of physical characteristics of volcanic eruptions that are
correlated with the size of resulting losses (in this case, height of the eruptive column and predominant direction of
ash dispersal), in order to pre-determine payments to the risk cedant concerned. We apply this risk transfer
mechanism to the case of Mount Fuji in Japan, by considering a potential risk cedant such as a regional Government
interested in offsetting losses to dwellings in the heavily populated Prefectures of Tokyo and Kanagawa. The
simplicity in determining eruptive column height and ash fall dispersal direction makes this design suitable for
extrapolation to other volcanic settings world-wide where significant ash fall producing eruptions may occur,
provided these parameters are reported by an official, reputable agency, and a suitable loss model is available for the
volcanoes of interest.




















## 1 Introduction

Volcanic eruptions are complex phenomena that generate a variety of hazards such as lava flows, ash fall, pyroclastic flows, lahars, and volcanic earthquakes. These may in turn cause physical damage to man-made structures and the discontinuation of activities related to aviation, tourism, and agriculture, among others.

Although rare, large volcanic eruptions pose significant destructive and disruptive potential. A medium-sized eruption like the 2010 Eyjafjallajökull eruption in Iceland (VEI[1] 4) caused the cancellation of about one hundred thousand flights and carried an estimated global cost of US$4.7 Billion (Oxford Economics, 2010). According to estimates by the Government of Japan, a repeat of the December 1707 Mt. Fuji eruption (VEI 5) could result in national losses over US$22.5 Billion (Cabinet Office of Japan, 2002), not including impacts on transportation and power transmission facilities that could effectively paralyze the Tokyo metropolitan area. Mt. Tambora's 1815 eruption in Indonesia (VEI 7) is regarded as the greatest eruption in historic time, ejecting as much as 175 km$^3$ of pyroclastic material that reached heights of over 40 km into the atmosphere (Self et al., 1984). It caused an estimated death toll of 71,000 people some of which due to the immediate explosion that killed around 12,000 people on Sumbawa Island (Oppenheimer, 2003). The event triggered tsunami waves striking several Indonesian islands and a famine related to eruptive fallout ruining crops in the region (Stothers, 1984; Oppenheimer, 2003). At present, over one million people live within 100km of Mt. Tambora (GVP, 2019).

Insurance is a mechanism to protect against financial losses from natural perils. Through insurance, people and entities transfer risks to insurance companies in return for the payment of an annual premium. These premiums are accumulated in order to build up reserves that enable them to pay claims in case of need. Insurance companies, similarly, can accept only a certain amount of risk, after which they may themselves seek protection through reinsurance. Companies who sell reinsurance are typically global in nature, hedging their risk in one region by selling products in another or by seeking insurance mechanisms themselves for their own portfolios (this is called "retrocession"). Through this chain of risk transfer accumulations of risk are successfully shared among many parties across the world, ideally enabling our society to cope with potentially large losses without any particular entity in this chain suffering unrecoverable losses.

As concentrations of risks grew, the capital available to supply global reinsurance products was in more demand, which had the consequence of raising prices. A larger supply of capital was necessary and there were large yields available for those interested. This gave rise to the appearance of Insurance Linked Securities (ILS), a type of financial instrument that allowed the capital markets to enter the insurance space in what has been referred to as "the convergence market," thus increasing the amount of capital available for insurance-related operations. One tool that falls into this category is a catastrophe (cat) bond, a means of fragmenting risk into coupon bonds that can be sold to qualified investors (Cummins, 2008; Swiss Re, 2011).

As new investors in this space lack familiarity with traditional insurance operations, there has been an interest in devising some of these instruments as a form of derivative that simplifies the process of settling a claim (World Economic Forum, 2008). This motivation gave rise to "parametric cat bonds" in which recoveries after a catastrophe event are tied to the occurrence of a set of measurable physical characteristics, such as the magnitude of an earthquake or the category of a hurricane, rather than to actual losses or indemnity. Properly chosen parameters that are easy to measure transparently and with accuracy can provide parametric cat bonds with a speed of payment unparalleled in the domain of insurance. Progressively, as sensors become more ubiquitous and precise, and as

---

[1] The Volcanic Explosivity Index (VEI) is a relative measure of the explosiveness of volcanic eruptions devised by Chris Newhall and Stephen Self in 1982. The scale is open-ended with the largest eruptions in history given magnitude 8. The scale is logarithmic from VEI 2 upwards, with each interval on the scale representing a tenfold increase in volume of eruptive products.



technology facilitates communication of measurements, parametric insurance mechanisms are becoming more
widespread.
Earthquake parametric cat bond transactions appeared first in 1997 and grew in number throughout the following
years, supported by what were then relatively novel techniques to model earthquake risk in the insurance market
(Franco, 2014). Since then, these earthquake solutions have taken many forms depending on the parameters chosen
for their design and on whether they are binary (pay or no pay) or "index-based" indicating a payment somewhat
correlated with the intensity of the event (Wald and Franco, 2016; 2017). A similar development in the field of
volcanic risks has not yet taken place. Only one product exists in the market, offered by Sompo Japan Nipponkoa
Insurance that provides coverage on a parametric basis for volcanic eruptions. This product is addressed to
commercial corporations in Japan at risk of experiencing losses derived from a volcanic eruption (Artemis, 2016).
Tailored in particular to the tourism industry, it grants coverage of losses up to US$10 million from business
interruption caused by the onset of a level 3 or above eruption alert as determined by the Japan Meteorological
Agency (JMA) (Yamasato et al., 2013).
The dearth of insurance derivative products linked to physical characteristics of volcanic eruptions may be partly
explained by the lack of fully probabilistic volcano loss models, which are a pre-requisite for the design and
calibration of these products. In this paper we present a stochastic volcanic risk model for six Japanese volcanoes on
which we base the construction of a parametric risk transfer tool. First, in Sect. 2 we describe the components of the
risk model; i.e. hazard, vulnerability, exposure, and loss computation. In Sect. 3, we discuss the conceptualization
and the mathematical design of a plausible parametric risk transfer tool leveraging physical descriptors of the
eruptive events that are both simulated in the risk model as well as reported by public entities during the course of an
actual event. The work draws from efforts carried out in the development of parametric triggers for other perils,
fundamentally earthquake (Franco, 2010; Franco, 2013; Goda, 2013; Goda, 2014; Pucciano et al. 2017; Franco et al.
2018) and tsunami (Goda et al. 2018). Sect. 4 applies the framework presented to an application case study in Japan
where a regional (or national) entity may desire to adopt this type of risk transfer mechanism to help offset costs
associated with ash-fall generated by an eruption of Mt. Fuji. Conclusions and final remarks are collected in Sect. 5
where we elaborate on the potential application of this type of tool in a generalized, volcanic, global setting.
**2    Construction of a volcano risk model**
Japan is one of the most volcanically active countries in the world. There are 111 active volcanoes in Japan; on
average, a total of 15 volcanic events (including eruptions) occur every year, some of which seriously hinder human
life (JMA, 2019). Five Japanese cities, Tokyo, Osaka, Nagoya, Sapporo and Fukuoka, are ranked among the top-20
cities most at risk from volcanic eruptions according to the Lloyd's City Risk Index (Lloyd's, 2018).
The development of a volcanic risk model for Japanese volcanoes allows improving our ability to quantify said risk
as a preliminary step to transferring it to the capital markets. The model focuses on physical damage of buildings
arising from significant deposition of volcanic ash (tephra). The geographic scope is limited to the highly populated
and industrialized Prefectures of Tokyo and Kanagawa, potentially affected by the surrounding six major volcanoes:
Fuji, Hakone, Asama, Haruna, Kita-Yatsugatake and Kusatsu-Shirane (see Fig. 1). The model presented does not
consider damage to contents, business interruption, or costs associated with ash fall clean up. Neither does it
consider other volcanic hazards such as lava flows, pyroclastic density currents, debris flows or avalanches. The
model is structured into four modules: hazard, vulnerability, built environment (or exposure), and loss calculation,
which are described in more detail in the following subsections.





**Figure 1: The geographic domain of the volcano ash fall model presented in this paper includes Tokyo and Kanagawa**
**Prefectures in Japan, and the six major volcanoes that can affect them, Fuji, Hakone, Asama, Haruna, Kita-Yatsugatake,**
**and Kusatsu-Shirane.**

*2.1 The hazard module*
The hazard module consists of a collection of 26,807 volcanic ash fall footprints, each of them associated with one
of the six modelled volcanoes and with an annual probability of occurrence (see Table 1).

**Table 1: Number of volcanic ash fall events included in the model (i.e. those ash fall events that impact the model's**
**geographical domain of Tokyo and Kanagawa prefectures) and associated annual probabilities of occurrence by volcano.**
**Ash fall events originated by these volcanoes that do not impact the model domain have been excluded from the counts.**
This original set of footprints was produced by Risk Frontiers in 2017, and was provided specifically for the purpose
of building the volcano risk model that we present in this paper, on an exclusive basis. Modelling was performed
using *tephra2* numerical model, which simulates the dispersion of ash fall from a volcanic source using mass
conservation and advection-diffusion equations (Bonadonna et al., 2005; Connor and Connor, 2006). Tephra
accumulation is computed for specified locations surrounding a volcano in load units ($Kg \times m^{-2}$). The model takes
into account appropriate vertical wind speed and direction profiles, which in this case were generated from
reanalysis wind data (NCEP-DOE Reanalysis2; NOAA).
The interaction of volcanic ash fall with rainfall may lead to an increase in the weight of the earlier due to absorption
of water, leading to increased loads and consequently to potentially more severe damages of affected structures. In
order to consider the possibility of ash fall – producing eruptions being concurrent to rainfall, "wet" versions of the
footprints were produced, respecting the rainfall patterns in the region of interest. The methodology used to create
"wet" footprints follows that described by Macedonio and Costa, 2012, and rainfall data were supplied by JBA Risk
Management. This was in the form of 10,000 years of simulated daily precipitation that incorporates tropical
cyclone and non-tropical cyclone precipitation; derived by JBA as part of their Global Flood Event Set.
*2.2 The vulnerability module*
As mentioned prior, the model considers damage to buildings only (residential, commercial or industrial), arising
from the vertical loads imposed by tephra on the structures. The level of damage to a specific building depends on
the total ash load and on the structural characteristics of the building. For each building type (i.e. a defined
combination of construction type, building rise and roof pitch) the model uses a specific vulnerability function that
computes the probability of experiencing a certain level of damage (expressed as a damage ratio of cost of repair
versus total cost of replacement) for a given physical load value upon that structure. The vulnerability functions
were developed on the basis of several studies on the subject (Spence et al.; 2005; Maqsood et al., 2014; Jenkins et
al., 2014; Jenkins et al., 2015; Blong et al., 2017) for building typologies common in the area (see Table 2). Given
the lack of data on roof type for individual structures, the model assumes probabilities of different roof types within
the exposure set (low, medium or high pitch) depending on the building occupancy, construction typology and
building rise.




**Table 2: Building types common in the Tokyo and Kanagawa Prefectures of Japan, for which specific vulnerability functions were developed in the volcano risk model. RC-SRC stands by "Reinforced Concrete – Steel Reinforced Concrete".**

Examples of damage functions used in the volcano risk model are provided in Fig. 2 for two contrasting building types (different construction type, building rise and roof pitch).

**Figure 2: Damage functions for two different building types considered in the volcano risk model ("RC-SRC" stands for Reinforced Concrete- Steel Reinforced Concrete; "Med." stands for Medium); source of these damage functions is Maqsood et al., 2014.**

*2.3 The exposure and the built environment (BE) modules*

These two closely-related modules jointly define the characteristics and monetary values of the group of buildings ("portfolio") for which the model will produce risk metrics.

1) The exposure module consists of a database structure that allows the user to characterise the portfolio of interest and upload those details to the risk model in a structured manner, to subsequently run it. The main database fields relate to number of buildings and associated values (i.e. building replacement values), geographical location of the buildings (supported geocoding levels include geographical coordinates, 5 and 7 digit Postal Codes and Prefecture), occupancy, construction type and building rise.

2) The BE module is a database that completes the information provided by the user, wherever it is incomplete or not accurate enough. This database represents the built environment across the model geographical domain, specifically, the number, characteristics and spatial distribution of the different building types as described in Table 2. The purpose of this module is two-fold. On one hand it allows defining the likely location of buildings geo-located at resolutions coarser than geographical coordinate, in order to better characterise their relationship with the spatial distribution of the hazard. The BE distributes buildings into a finer spatial resolution on a probabilistic basis, using weights that are specific to each building type. Weights were computed on the basis of information such as land use and land cover type and census data. In the case of our model, data sources included the 2013 Housing and Land Survey (Statistics Bureau, Government of Japan), the 2014 Tokyo Statistical Yearbook (Tokyo Metropolitan Government), Japan E-Stat (Ministry of Land, Infrastructure, Transport and Tourism), etc. The second purpose of the BE is to infer damage-relevant characteristics of buildings (e.g. building rise, construction type, etc.) if this information is not captured in the description of the buildings we want to model. This is again done on a probabilistic basis, depending on the location of the building and any known characteristics (e.g. building occupancy).

*2.4 The loss calculation module*

The loss calculation module or engine estimates the monetary loss associated to each building for the different events that can potentially affect it. This is attained (for each event-building "interaction") by multiplying the damage ratio prescribed by the corresponding vulnerability function and the replacement value of the building, which needs to be provided by the modeller. The loss calculation module allows reporting losses by building and by event; as well as by event (aggregate event loss).





Volcanic loss data are very scarce due to the low frequencies of damaging eruptions. We used a few independent
sources to validate modelled losses. These included two studies on damage estimations of a repeat of the 1707 Fuji
eruption (Kuge et al., 2016; Cabinet Office of Japan, 2002) that were used to validate modelled losses from severe
eruptions. To validate modelled losses from less severe eruptions, we used as a proxy data on insured building losses
caused by loading of snow in Toyo and nearby Prefectures in February 2014 (General Insurance Association of
Japan, 2015). Kuge et al. (2016) modelled losses for industrial buildings (with an assumed value of 1 Billion JPY
per building) if there was a repeat of the Fuji 1707 eruption. Estimated individual building losses ranged between 35
and 180 Million JPY (K. Kuge, personal communication, 2017). This compares well with our modelled losses
between 28.6 and 138.4 Million JPY for industrial buildings, under a reconstruction of the Fuji 1707 eruption.
Regarding Residential buildings, the reported average building loss value for the February 2014 snowfall event in
Japan was 1.2 Million JPY (General Insurance Association of Japan, 2015). Assuming a snow density value of 200
$kg/m^3$, we identified ash fall events in the volcano model producing equivalent loads, and calculated an average
Residential building loss of 1.7 Million JPY.

## 3    Design of a parametric trigger for volcano risk transfer

A parametric trigger refers to a specific value or threshold of a physical, measurable characteristic associated to the
natural phenomenon in question (e.g. to ash fall-producing volcanic eruptions in this case, or earthquakes,
hurricanes, etc.), above which a significant level of damage of exposed assets (e.g. damage to buildings) is likely to
occur. When the physical parameter exceeds that threshold for a particular event, it is considered that a risk cedant
should receive a payment commensurate to the loss that their portfolio will likely incur as a result of being exposed
to the event.
Therefore, when designing a parametric risk transfer mechanism, it is crucial to select a physical parameter that
correlates well with potential losses. In the case of parametric earthquake risk transfer, for instance, it is common to
select the magnitude of the earthquake as the main parameter, and subsequently define threshold value/s for the
magnitude scale, above which significant damages are likely to occur (Franco, 2010; Franco, 2013). Other
alternatives used in practice consider shaking measurements such as peak ground accelerations or spectral
accelerations at a set of locations (Goda, 2013; Goda, 2014; Pucciano et al. 2017).
There are three important requirements for the selection of a physical characteristic of a natural phenomenon to be
used as a parametric trigger in the design of a risk transfer mechanism:
1)   The parameter must exhibit strong correlation to losses incurred as a consequence of the physical phenomenon.
2)   The parameter needs to be measured and reported by a reliable and impartial organisation on a near-real time
basis. In the case of earthquakes, for instance, earthquake information is often obtained from reliable
international bodies such as the U.S. Geological Survey (Wald & Franco, 2017).
3)   Finally, each of the stochastic events in the catastrophe risk model used as a basis to design the risk transfer
solution must explicitly include the corresponding value for the selected physical parameter. In the case of
earthquake risk transfer, for instance, each of the earthquake events in the catastrophe risk model needs to be
described by its magnitude (if this is the metric of choice for the trigger conditions).

### 3.1 Choosing the trigger parameters for volcanic eruptions

In our case study, we have researched several physical parameters associated to the phenomenon of volcanic ash
falls, as well as Japanese organizations reporting this type of information on a real-time basis while a volcanic
eruption unfolds. In Japan, the Japanese Meteorological Agency (JMA) operationally monitors volcanic activity
throughout the country and issues relevant warnings and information to mitigate related damages. To continuously
monitor volcanic activity, JMA deploys seismographs and related observation instruments in the vicinity of 50





volcanoes that are remarkably active in Japan. When volcanic anomalies are detected, the Agency steps up its
monitoring/observation activities and publishes volcanic information and regular bulletins; mainly "Observation
Reports on Eruption" and "Volcanic Ash Fall Forecasts" (VAFFs). The Observation Reports and VAFFs are
published on a real-time basis for all active volcanoes in Japan; however they contain different types of information.
Observation Reports provide information on the ongoing eruption, such as eruption time, eruptive column height (in
meters above the crater), the main direction of movement of the eruptive plume at the moment of the report (as per
eight cardinal directions: N, E, SE, etc.…), and the maximum plume height recorded from the onset of the eruption
(Hasegawa et al., 2015). On the other hand, the VAFFs consist of modelled (not observed) ash fall areas and
amounts, and are produced when heavy (> 1 mm) or moderate (0.1-1 mm) ash quantities are forecasted in principle.
These maps correspond to the moment when the VAFF is issued, and cumulative ash fall map products (i.e. the total
accumulated ash fall on the ground throughout the eruption) are not released by JMA.
Eruptive column height values are available for each eruptive event present in the volcano risk model. In addition,
we estimate the predominant direction of movement of the eruptive plume for each event by assuming it coincides
with the main axis of ash fall deposition on the ground. Therefore, we calculate the main direction of deposition of
ash fall for each of the event footprints in the model by performing spatial analyses. Resulting azimuths were
classified into eight directional sectors (N, NE, E, SE, S, SW, W, and NW) and used as a proxy for the main
direction of movement of the generating eruptive ash plume.
Based on the above, we selected a combination of two eruption-related parameters (reported eruptive column height
and direction of movement of the eruptive plume) for the design of our parametric trigger, since:

1)  These two parameters are reported by JMA on a near-real time basis when an eruption occurs.
2)  The height of the eruptive column and preferential direction of movement of eruptive plume for each of the
stochastic events in the model can be assigned based on existing datasets.
3)  We found a significant relationship between eruptive column height and losses as modelled by the volcano
risk model (Fig. 3). Pearson correlation tests were performed between eruptive column height and losses,
for eight subsets of eruptive events with defined eruptive plume directions (i.e. E, N, NE, NW, S, SE, SW,
295            W). Resulting p-values were all smaller than alpha = 0.05, indicating a significant correlation between
eruptive column height and losses for all directional sectors.

We do not consider modelled ash fall areas for the parametric design, given that cumulative maps are not made
typically available and it is thus not straightforward to establish a relationship with losses.

**Figure 3: Relationship between height of eruptive column (in Km, from crater rim) and modelled losses for all eruptive**
**events in the volcano risk model. Each panel displays a subset of eruptions featuring a specific predominant direction of**
**their eruptive plume (East, North, North-East, North-West, South, South-East, South-West and West).**

**3.2 Choosing the trigger type**
The next step consists of designing the parametric trigger on the basis of the two physical eruptive parameters
selected. We have however, several choices in the formulation of such a trigger (Wald & Franco, 2016; Pucciano et
al., 2017). In this paper, we focus on two simple variants:

1)  *Binary triggers*, for which each event of the stochastic catalogue can either pay or not pay a fixed monetary
amount, *P*, depending on whether it exceeds the parameter threshold defined by the specific design.





2) *Multilayer triggers*, for which each event can pay one of $N$ predefined payment levels, associated to a series of defined parameter thresholds.

The binary trigger can be seen as a particular case of a multilayer trigger with $N = 1$. As treatment of this case is easier, we start with the design of a binary parametric trigger and we later generalize it to $N$ payment levels.

Since we are building a trigger using plume height and ash plume direction expressed as per eight wind sectors (N, NE, E, SE, S, SW, W, NW), it is natural to represent the trigger simply as a set of threshold plume height values for each wind sector, $\{H_s\}_{s \in W}$, where $W$ is the set of the possible wind sectors.

This means that if an event $i$ has plume height $h_i$ and wind sector $s_i$, it triggers a payment if and only if $h_i \geq H_{s=s_i}$, which is the *trigger condition*.

We can model the behaviour of the trigger using the stochastic events in the volcano risk model. Let's call $T$ the set of the stochastic events fulfilling the trigger conditions. Since they are the only events releasing a payment, their exceedance rate, collectively, defines the payment occurrence rate.

$$R = \sum_{i \in T} r_i$$

where $r_i$ stands for the event occurrence rate. From the trigger rate we obtain the yearly triggering probability as $p = 1 - e^{-R}$ as usual for a Poisson process. The expected payment in a year can be expressed either as $EP = p \cdot P$ or $EP = R \cdot P$ but since we generally have $p \sim R$ the impact of the difference is minimal.

If we interpret the trigger as insurance, the $EP$ would correspond to the *pure premium* of the policy, which is a quantity somewhat proportional to its price. Thus, the more often the trigger is activated the more expensive it is. Given a certain trigger payment and a certain yearly budget, we can thus derive a target triggering rate $R^*$.

Since the trigger pays a fixed amount, it will always provide either too much money or too little, if compared to the actual event loss. This difference is expressed via the following quantity, called **basis risk,** which we define based on Franco (2010) as:

$$BR = BR_+ - BR_- = \sum_{i: l_i < P} (P_i - l'_i)\, r_i \; - \sum_{i: l_i > P_i} (l'_i - P_i)\, r_i$$

Where $P_i = P$ if $i \in T$ and $0$ otherwise and $l'_i$ represent the loss component in the loss layer of interest. The first (second) term is called positive (negative) basis risk.

**3.3 Optimization of the trigger**

The standard approach to trigger design consists of choosing the trigger thresholds such that basis risk is minimized (Franco, 2010; Goda, 2013; Goda, 2014; Pucciano et al., 2017). Since the budget and the trigger recovery do tend to change during the design process, recent approaches have considered the alternative objective that the trigger simply maximizes the amount of **risk transfer** (Franco et al., 2018; Franco et al., 2019), i.e. find $T$ that maximizes the quantity defined as:

$$K = \sum_{i \in T} r_i l_i$$





Where $l_i$ is the loss for event $i$, that is, we want a trigger which is activated by those events in the catalogue that
collectively have the greater expected annual loss. Maximizing the risk transfer is quite apt, since it states clearly
that the trigger is designed to be activated on the set of events that affect the policy holder the most.
Using the trigger condition we can rewrite the risk transfer equation in function of the trigger parameters as

$$K(\{H_s\}_{s\in W}) = \sum_{s\in W} \rho_s(H_s) = \sum_{s\in W} \sum_{i:\,h_i\geq H_{s=s_i}} r_i l_i \quad (1)$$

Where $\rho_s(H_s)$ is the risk transferred by all the events in sector $s$, which is a function of the threshold value for that
sector, $H_s$.
If we discretize the possible values of $H_s$ in a vector, $H_s^k$, and we compute all the possible values of $rt_s$ for this
vector, $\rho_s^k = \rho_s(H_s^k)$, we can rewrite the risk transferred per sector as

$$\rho_s(H_s) = \sum_k x_s^k \rho_s^k \quad (2)$$

Where $x_s^k$ is a vector of 0 and one single 1 placed at the index $k'$ such that $H_s^{k'} = H_s$. This means that we can write
$H_s$ as

$$H_s = \sum_k x_s^k H_s^k$$

When plugging Eq. (2) in Eq. (1), the risk transfer equation becomes

$$K = \sum_{s\in W} \sum_k x_s^k \rho_s^k$$

It seems an over complication of a previously simple equation, but actually we eliminated the sum over $i \in T$. Now
the unknown is moved from the set $T$ to the vectors $x_s$ which resembles a problem of linear algebra (it's not, given
the particular form of the vectors, but it's still easier to approach than before). We can now apply similar
considerations to the rate equation obtaining an expression for the payment occurrence rate

$$R = \sum_{s\in W} \sum_k x_s^k \lambda_s^k$$

where $\lambda_s^k = \sum_{i:\,h_i\geq H_{s=s_i}^k} r_i$. At this point we can re-write the trigger design as the following optimization problem:

$$\text{find the } x_s^k$$

$$\text{which maximize } \sum_{s\in W} \sum_k x_s^k \rho_s^k$$

$$\text{subject to the following constraints:}$$

$$\sum_{s\in W} \sum_k x_s^k \lambda_s^k \leq R^*$$

$$\sum_k x_s^k H_s^k - \sum_k x_{s'}^k H_{s'}^k \leq \Delta H \quad \forall \text{ adjacent } s, s'$$





$$\sum_k x_s^k = 1 \quad \forall s$$

$$x_s^k \in \{0, 1\}$$

Where $R^*$ is the target trigger rate and $\Delta H$ is a maximum threshold difference between two adjacent wind sectors.
Limiting this difference is a way to take into account epistemic risk, that is, risk induced by using a particular model.
It is also a way to decrease trigger sensitivity to the wind sector parameter.
The last two constrains, instead, are just a way to express the peculiar form of the $x_s$ vectors.
The problem, thus stated, can be solved with linear programming techniques (Franco et al., 2019) or with other
alternative methods (De Armas et al., 2016). The problem is solved in this paper using standard Python libraries for
mixed integer linear programming.
As can be seen from the equations for $K$ and $R$ , these two quantities are non-decreasing when the number of trigger
events increases. Thus, maximizing $K$ involves increasing the number of events captured by the trigger (by
decreasing the threshold values) up to a certain point where the critical value $R^*$ is reached. This constraint, as all
the other constraints of the optimization, imposes a trade-off to the $\max(K)$. The curve described by $\max(K)$ in
function of $R^*$ is a Pareto front, an example of which is depicted in Fig. 4.

**Figure 4: Pareto front for a binary trigger designed modelling stochastic losses for Mt. Fuji. The transferred risk is**
**displayed as percentage of the total risk.**

In a multi-layer payment trigger, instead of having one single threshold height value we have a series of threshold
values for each wind sector. Each threshold value pays a certain fraction of the maximum payment. Let's suppose
we can generate a two-layer trigger. We decide in advance that the occurrence rate of the first and second payment
will be $R_1^*$ and $R_2^*$ respectively, with $R_1^* > R_2^*$.
To build the trigger we follow these steps.
1)   We build a binary trigger, $\left\{H_s^{(1)}\right\}_{s \in W}$, with occurrence rate $R_1^*$
2)   We build a second trigger with occurrence rate $R_2^*$. The problem is identical to the binary one, but with an
additional constraint:

$$\sum_k x_s^k H_s^k > H_s^{(1)} \quad \forall s$$

Which means that each threshold must be greater or equal to the threshold for that sector in the lower layer. It is easy
to generalise to $N$ layers imposing at each layer $n$ the constraint $H_s^{(n)} > H_s^{(n-1)} \quad \forall s$.

**4   Application and Results**
For this application, we consider a case where a cedant such as a regional Government may want to consider
financing the risk of economic losses arising from damage to citizens' residential properties in the Prefectures of
Tokyo and Kanagawa, caused by the potential occurrence of damaging eruptive ash fall events. We assume that the

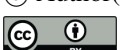



Government has an implicit need to help reconstruct citizens' dwellings after a catastrophic volcanic event, and may
therefore want to consider adopting a parametric risk transfer solution appropriately designed for these cases.
The first step consisted of putting together a comprehensive "portfolio" of residential properties for the modelled
geographical area (Tokyo and Kanagawa Prefectures). This portfolio is the input that needs to be provided to the
volcano risk model, for it to calculate potential losses on a probabilistic basis. To do so, we used the census data
incorporated in the model database, which consists of the number of dwellings by administrative unit (Shiku) and by
type of residential occupancy (single family or condominium). The cost of rebuilding each of the properties also
needs to be provided to the model, and we used different information sources to estimate representative rebuilding
costs for single family dwellings and condominiums in the prefectures of Tokyo and Kanagawa (Table 3).

**Table 3: Representative reconstruction values have been estimated on the basis of several sources of information,**
**including data on building construction values from Japanese Government Statistics (https://www.e-stat.go.jp) and**
**insured building values from the General Insurance Rating Organization of Japan (https://www.giroj.or.jp).**

Table 4 provides a summary of the total number of dwellings and corresponding total reconstruction values for the
modelled portfolio.

**Table 4: Total number of dwellings and total reconstruction values modelled in the volcano risk model for six Japanese**
**volcanoes (by prefecture, and totals). Number of dwellings from Japanese Government Statistics (https://www.e-**
**stat.go.jp); Total Values have been calculated on the basis of representative reconstruction values in Table 3.**

The volcano risk model was run and results were extracted as an "Event Loss Table" or "ELT" (i.e. losses produced
by each of the volcanic ash fall events included the model, on the residential portfolio considered). Table 5 provides
an example of results for a few ash fall events from Mt. Fuji. Losses can be equal to zero for events either impacting
areas outside the model's geographical domain (i.e. Tokyo and Kanagawa prefectures), or impacting geographical
areas within the model domain that have no (modelled) buildings located in them.

**Table 5: Subset of ELT outputs from the volcano risk model, run of the residential portfolio described. The table shows**
**losses on the portfolio caused by four of the model's ash fall events from Mt. Fuji. The mean loss and the standard**
**deviation of the loss distribution associated to each event (in JPY) are reported in the ELT.**

The ELT results were used to analyse the correlation between height of eruptive column and modelled event losses
(Fig. 3), which is a pre-requisite for the selection of this metric for the design of the parametric trigger. Figure 3
plots, for each modelled ash fall event, the height of the eruptive plume (x axis) versus the logarithm of the modelled
loss (y axis), showing a strong correlation between the two. Each panel in Fig. 3 depicts eruptive events featuring a
specific predominant dispersal direction of their eruptive plume (East, North, North-East, North-West, South, South-
East, South-West and West). The correlation between plume height and loss holds for all direction sectors.
Dispersion in the plot is due to the fact that the severity of loss, despite being strongly correlated with plume height
and plume direction, also depends on other factors, such as duration of the eruption, size distribution of eruptive
particles, etc.


Calculation of Annual Average Losses (AAL) for the modelled portfolio on a per-volcano basis (Fig. 5, left) shows
that Mont Fuji is the main risk source, its average AAL amounting to more than 1 billion JPY per year. Therefore,
we chose Mt. Fuji for the calculation of the parametric risk transfer structure. Being located westward of the
exposure domain, risk associated to Mt. Fuji is mainly concentrated in the eastern wind sector. In particular, the only
sectors containing risk are NE, E, SE, S and SW, even if the last three only in minimal part (Fig. 5, right).

**Figure 5: (Left) Modelled AAL for the six volcanoes included in the volcano risk model. (Right) Breakdown of Mt Fuji**
**risk by wind sector.**

The occurrence exceeding probability curve (OEP) derived from the modelled losses for Mt. Fuji is depicted in Fig.
6. As an example, we imagine that the policy holder might be interested in covering all losses exceeding 30 Billion
JPY with a parametric coverage releasing two possible payment levels of 100 and 300 Billion JPY. This means

$$l_i' = \min(\max(l_i - 30\text{B}, 0), 300\text{B})$$

We choose the target exceedance rates for these layers to match the corresponding return period on the OEP curve,
3862 and 4944 years. In this way we end up with the trigger OEP curve depicted in Fig. 6.
We also imposed a plume height discretization of 1Km, i.e. $H_s^k = (1\text{Km}, 2\text{Km}, ....50\text{km})$ and a maximum threshold
difference between adjacent sectors $\Delta H = 4\text{Km}$.

**Figure 6: OEP curve for Mt Fuji losses (blue) and trigger payments (orange)**

The result of the optimization algorithm is depicted in Fig. 7. The (wind sector, plume height) plane is divided into
three payment regions, separated by the two trigger layers. As expected, the plume height thresholds are smaller for
regions of high risk. The smoothing condition ensures that there is coverage also in the sectors that are adjacent to
the sectors at risk, in case that an event has ash fall direction close to the border between two sectors and it is
categorized wrongly.

**Figure 7: Parametric Trigger for Mt. Fuji Each dashed line correspond to a unit of 10Km**

Table 6 summarizes the results of the parametric trigger design for the considered cover, including the plume height
thresholds by wind sector for the two Layers defined, and the corresponding proportion of risk transferred and layer
payments.

**Table 6: Parametric trigger for Mt Fuji. The risk transferred by each layer is expressed as percentage over the total risk**
**of Mt Fuji. The layer payment is expressed as fraction of the maximum payment (300 Billion JPY).**





The net basis risk of the trigger is 7 Million JPY / year, sum of 32 Million JPY / year of positive and 25 Million JPY / year of negative basis risk, while the expected recovery is of 87 Million JPY / year. The prevalence of basis risk is expected, since the OEP curve of the bond sits on top of the losses OEP in the layer of interest (30 Billion – 330 Billion JPY). This amount can be fine-tuned increasing the return periods of the layers until comfortable levels of basis risk are reached.

## 5 Conclusions

We present a novel methodology to parameterize financial risk transfer instruments for explosive, tephra fall-producing volcanic eruptions. The design of the parametric product relies on easily obtainable, observable physical parameters relating to explosive volcanic eruptions; namely maximum observed height of the eruptive column and the prevalent direction of dispersal of the associated ash plume.

We take as a case study Mount Fuji in Japan, the largest and closest active volcano to the populous Tokyo metropolitan area and the heavily industrialized Kanagawa prefecture (Yamamoto & Nakada, 2015).  In Japan, the JMA reports height of the eruptive column and the predominant direction of ash dispersal as part of the "Observation Reports on Eruption" that are released for any erupting volcano on a near-real time basis. The design of the parametric risk transfer for our case study relies on Guy Carpenter's fully probabilistic model for volcanic eruptions potentially affecting Tokyo and Kanagawa prefectures, which includes 10,000 simulated volcanic ash fall events arising from explosive eruptions of different sizes at Mount Fuji. Therefore, the second pre-requisite for the successful design of an equivalent parametric product elsewhere is the existence of a fully probabilistic eruptive loss model encompassing the range of all possible eruptive events of interest, and incorporating information relating to plume height and predominant direction of ash fall dispersal for each event.

For the parametric design, we focused on explosive eruptions producing significant tephra loads capable of generating property damages (these are the type of eruptive events considered by the volcano risk model). The resulting parametric product could be of interest to a number of organizations, including regional and national Governments, but also to economic sectors such as insurers of commercial and industrial properties in these Prefectures (insurance cover for volcanic eruptions is included as part of the standard earthquake policies in Japan). In our case study, we took as an example a "portfolio" of residential properties representing the existing residential building stock in the Tokyo and Kanagawa prefectures. These could be severely affected by a significant eruption from Mount Fuji- the last Fuji eruption in year 1707 is a good example - thus potentially generating a financial burden for the regional and/or or national Governments.

We designed a multi-layer trigger assuming that a policy holder might be interested in covering all losses exceeding 30 Billion JPY, with a coverage releasing two possible payment levels of 100 and 300 Billion JPY provided the appropriate trigger conditions of eruptive column height and predominant plume direction are met (Table 6). This product would provide a policy holder such as a regional Government a quick way to access cash to help repair damages incurred by dwellings as a consequence of a major volcanic eruption.

Further work on the design of volcano-related parametric risk transfer products may relate to different aspects. On one hand, and also considering ash fall-producing volcanic eruptions, the design may be extended to consider other types of damages such as those to crops and livestock, costs arising from ash fall clean up and disposal in urban areas and roads, Business Interruption costs arising from air traffic disruption, airport closures and disruption of critical infrastructures including transportation networks, electricity, water supplies and telecommunications, etc. (Wilson et al., 2012). For any of these types of losses, specific ash fall vulnerability functions must be incorporated in the fully probabilistic volcano model considered. The parametric design presented in this paper could be adapted to coverage of these types of losses, provided a strong correlation was also found between eruptive column height and main direction of ash dispersal and modelled losses.





On the other hand, despite ash fall is the volcanic peril with the largest potential of causing wide spread losses (since
it is by far the most widely distributed eruptive product), there are other volcanic perils that have a large destructive
potential, albeit with a more constrained spatial reach. These include lava flows, pyroclastic density currents, lahars,
volcano flank collapses and ballistic blocks. Design of parametric transfer products for these volcano hazards would
entail a rather different approach; concerning both the modelling of losses (starting with the incorporation of these
specific hazard events to the fully probabilistic volcano model), to the selection and monitoring of hazard-related
trigger parameters.
There are several features of the design presented that make it potentially applicable to other volcanic settings where
explosive volcanism is typical. In particular, the choice of eruption-related parameters (height of eruptive column
and preferential direction of dispersal of ash fall) means that no special monitoring equipment is needed for
recordings. On the other hand, it is important that an official, reputable national or regional agency reports such
observations in a reliable and timely manner. Implementation should be straight forward in countries with
established volcano observatories. In others, it could be interesting to explore global monitoring solutions like
satellite-based remote sensing to report both column height and preferential direction of ash fall dispersal on a near
real time basis. Such arrangement would provide for a centralised, consistent and independent monitoring solution
applicable to explosive eruptions world-wide.
The other important requisite that needs to be in place is a suitable volcano risk model that produces stochastic loss
outputs associated to ash fall-producing eruptions. In an insurance context, availability of such models is still rare.
Nonetheless, increased collaboration between academic experts and the insurance industry brings all the necessary
elements together for the creation of such models, as it has been in the case presented in this paper. Whereas
building of volcano loss models requires from a non-negligible investment of time and resources, the availability of
open-source hazard simulation models such as tephra2 and of global open databases (e.g. wind data, eruptive data,
etc.) means that the ingredients needed for development are pretty much available on a world-wide basis. Scaling up
such approach in order to model a significantly larger number of volcanoes than presented in this paper is currently
being looked into, with promising preliminary results. Increased interest in parametric risk transfer products from
the insurance industry and capital markets is helping build momentum for the development of risk models of "non-
traditional" perils such as volcanic eruptions, and the design of associated risk transfer mechanisms.















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



**Tables**

| Volcano Name | Number of ash fall events | Aggregate Annual Occurrence Probability |
|---|---|---|
| Fuji | 9,969 | $4.84 \times 10^{-3}$ |
| Hakone | 12,821 | $6.58 \times 10^{-4}$ |
| Asama | 832 | $8.45 \times 10^{-5}$ |
| Haruna | 651 | $3.95 \times 10^{-5}$ |
| Kita-Yatsugatake | 2,065 | $2.57 \times 10^{-6}$ |
| Kusatsu-Shirane | 469 | $6.01 \times 10^{-6}$ |

**Table 1: Number of volcanic ash fall events included in the model (i.e. those ash fall events that impact the model's geographical domain of Tokyo and Kanagawa prefectures) and associated annual probabilities of occurrence by volcano. Ash fall events originated by these volcanoes that do not impact the model domain have been excluded from the counts.**






| Function ID | Occupancy | Construction Type | Building Rise | Roof Pitch |
|---|---|---|---|---|
| 1 | Resid., Comm. or Indust. Buildings | Wood Frame | Low | Medium |
| 2 | Resid., Comm. or Indust. Buildings | Wood Frame | Low | High |
| 3 | Resid., Comm. or Indust. Buildings | Wood Frame | Medium | Medium |
| 4 | Resid., Comm. or Indust. Buildings | Wood Frame | Medium | High |
| 5 | Resid., Comm. or Indust. Buildings | RC-SRC or Steel Frame | Low | Low-Medium |
| 6 | Resid., Comm. or Indust. Buildings | RC-SRC or Steel Frame | Low | High |
| 7 | Resid., Comm. or Indust. Buildings | RC-SRC or Steel Frame | Medium | Low-Medium |
| 8 | Resid., Comm. or Indust. Buildings | RC-SRC or Steel Frame | Medium | High |
| 9 | Resid., Comm. or Indust. Buildings | RC-SRC or Steel Frame | High | Low-Medium or High |
| 10 | Resid. Buildings | Light Metal Frame | Low | Medium |
| 11 | Resid. Buildings | Light Metal Frame | Low | High |
| 12 | Resid., Comm. or Indust. Buildings | Light Metal Frame | Medium | Medium |
| 13 | Resid., Comm. or Indust. Buildings | Light Metal Frame | Medium | High |
| 14 | Resid., Comm. or Indust. Buildings | Light Metal Frame | High | Medium |
| 15 | Resid., Comm. or Indust. Buildings | Light Metal Frame | High | High |
| 16 | Comm. or Indust. Buildings | Steel Frame or Light Metal Frame | Low | Low-Medium; long-span |


**Table 2: Building types common in the Tokyo and Kanagawa Prefectures of Japan, for which specific vulnerability functions were developed in the volcano risk model. RC-SRC stands by "Reinforced Concrete – Steel Reinforced Concrete".**





| Prefecture | Type of Residential Dwelling | Representative reconstruction values (Million JPY) |
|---|---|---|
| Tokyo | Single Family | 25.5 |
| | Condominium | 16.3 |
| Kanagawa | Single Family | 22.1 |
| | Condominium | 12.3 |


**Table 3: Representative reconstruction values have been estimated on the basis of several sources of information, including data on building construction values from Japanese Government Statistics (https://www.e-stat.go.jp) and insured building values from the General Insurance Rating Organization of Japan (https://www.giroj.or.jp).**






























|  | Number of dwellings | Total Value (Million JPY) |
|---|---|---|
| **Tokyo** | 6,435,994 | 121,605,115 |
| **Kanagawa** | 3,828,279 | 62,788,449 |
| **TOTAL** | 10,264,273 | 184,393,564 |


**Table 4: Total number of dwellings and total reconstruction values modelled in the volcano risk model for six Japanese**
**volcanoes (by prefecture, and totals). Number of dwellings from Japanese Government Statistics (https://www.e-**
**stat.go.jp); Total Values have been calculated on the basis of representative reconstruction values in Table 3.**





| EventID | Volcano | Annual Event Rate | Mean Loss (JPY) | Loss S. Dev. (JPY) (Independent) | Loss S. Dev. (JPY) (Correlated) |
|---|---|---|---|---|---|
| 1588 | Fuji | $9.84 \times 10^{-8}$ | $1.03 \times 10^{12}$ | $1.28 \times 10^{9}$ | $1.32 \times 10^{11}$ |
| 1589 | Fuji | $3.65 \times 10^{-7}$ | $1.87 \times 10^{6}$ | $2.25 \times 10^{6}$ | $1.93 \times 10^{7}$ |
| 1590 | Fuji | $4.91 \times 10^{-8}$ | $1.36 \times 10^{13}$ | $4.29 \times 10^{9}$ | $1.01 \times 10^{12}$ |
| 1591 | Fuji | $9.82 \times 10^{-7}$ | 0 | 0 | 0 |


**Table 5: Subset of ELT outputs from the volcano risk model, run of the residential portfolio described. The table shows**
**losses on the portfolio caused by four of the model's ash fall events from Mt. Fuji. The mean loss and the standard**
**deviation of the loss distribution associated to each event (in JPY) are reported in the ELT.**




| | Plume Height Thresholds [Km] | | | | | | | | Yearly Exceedance Probability | Transferred Risk | Layer Payment |
|---|---|---|---|---|---|---|---|---|---|---|---|
| | N | NE | E | SE | S | SW | W | NW | | | |
| Layer 1 | 32 | 28 | 28 | 32 | 36 | 37 | 40 | 36 | 0.026% | 76% | 33% |
| Layer 2 | 33 | 32 | 29 | 33 | 37 | 40 | 41 | 37 | 0.020% | 67% | 100% |



**Table 6: Parametric trigger for Mt Fuji. The risk transferred by each layer is expressed as percentage over the total risk of Mt Fuji. The layer payment is expressed as fraction of the maximum payment (300 Billion JPY).**






**Figures**




**Figure 1: The geographic domain of the volcano ash fall model presented in this paper includes Tokyo and Kanagawa**
**Prefectures in Japan, and the six major volcanoes that can affect them, Fuji, Hakone, Asama, Haruna, Kita-Yatsugatake,**
**and Kusatsu-Shirane.**



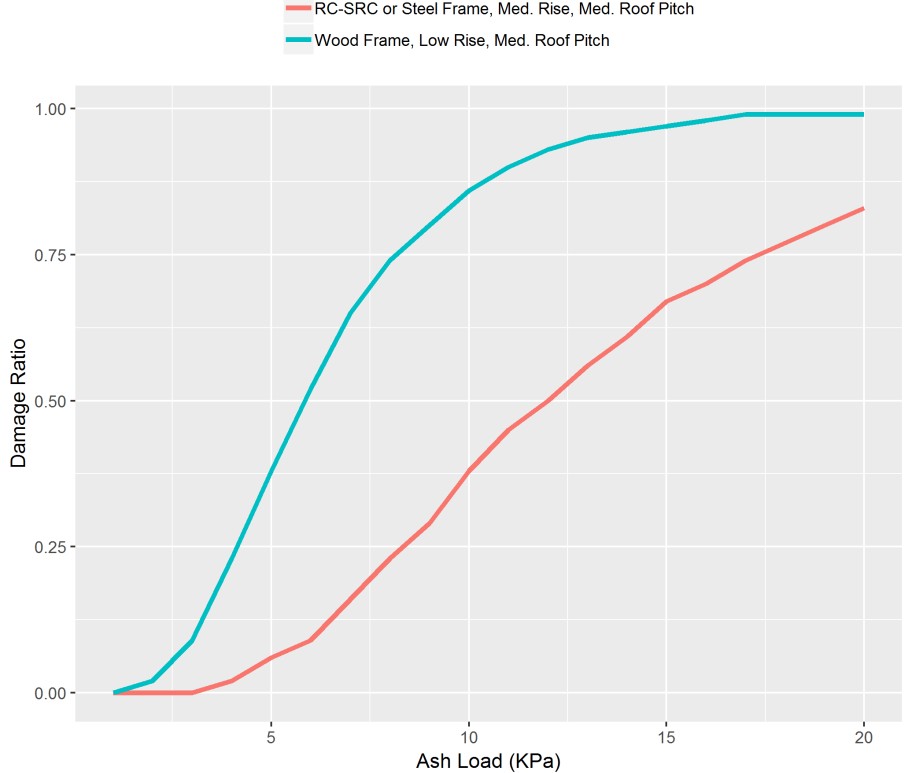

**Figure 2: Damage functions for two different building types considered in the volcano risk model ("RC-SRC" stands for**
**Reinforced Concrete- Steel Reinforced Concrete; "Med." stands for Medium); source of these damage functions is**
**Maqsood et al., 2014.**





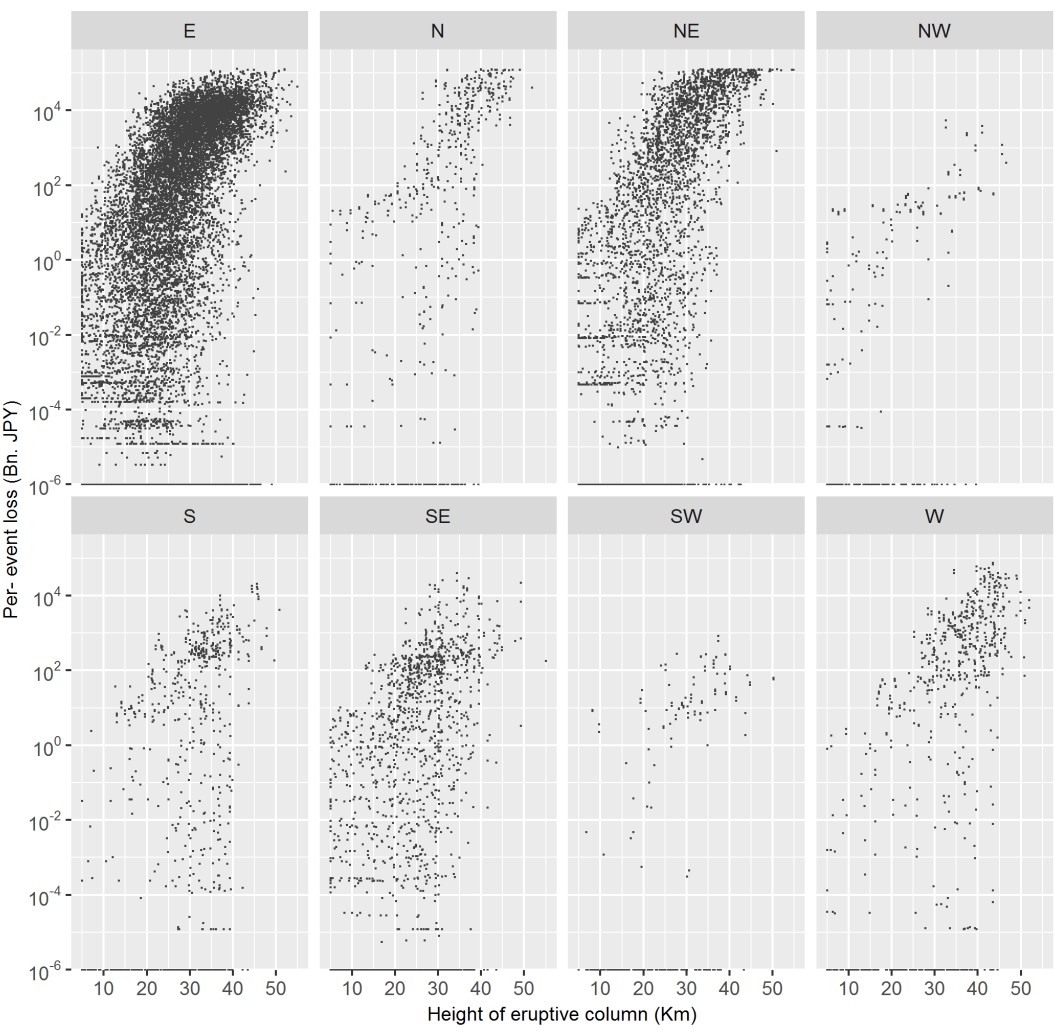


**Figure 3: Relationship between height of eruptive column (in Km, from crater rim) and modelled losses for all eruptive events in the volcano risk model. Each panel displays a subset of eruptions featuring a specific predominant direction of their eruptive plume (East, North, North-East, North-West, South, South-East, South-West and West).**






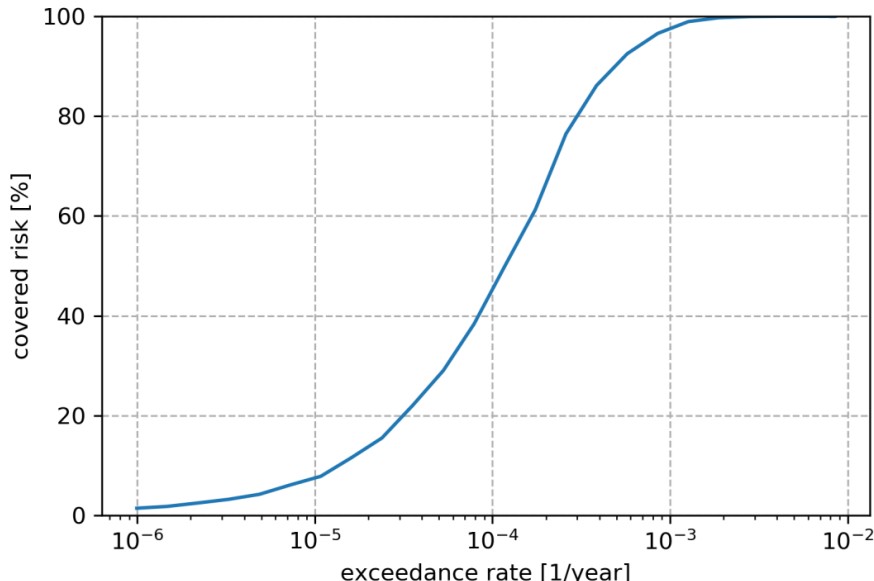


**Figure 4: Pareto front for a binary trigger designed modelling stochastic losses for Mt. Fuji. The transferred risk is displayed as percentage of the total risk.**







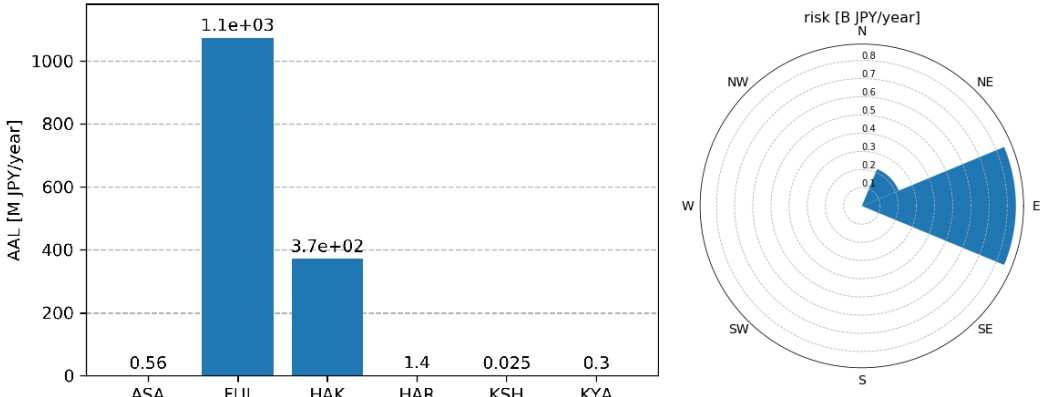

**Figure 5: (Left) Modelled AAL for the six volcanoes included in the volcano risk model. (Right) Breakdown of Mt Fuji**
**risk by wind sector.**





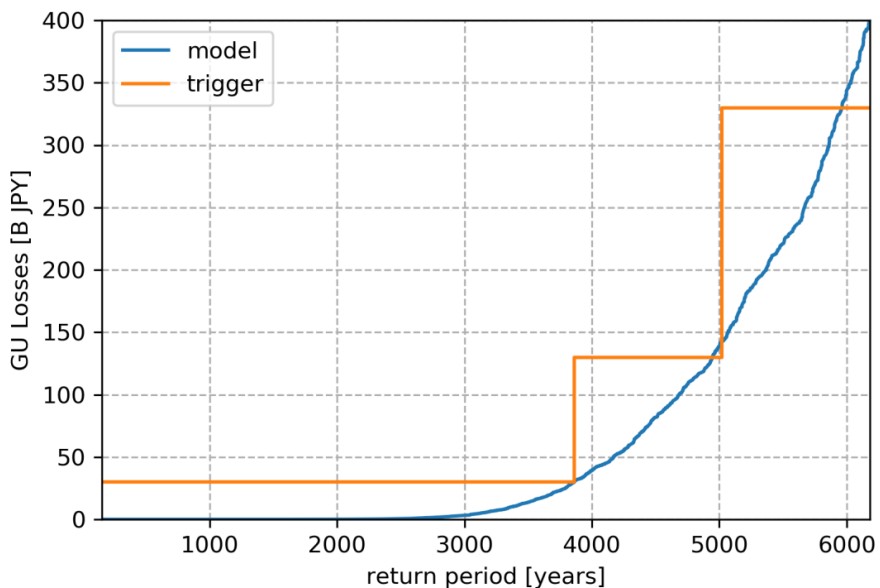

**Figure 6: OEP curve for Mt Fuji losses (blue) and trigger payments (orange)**






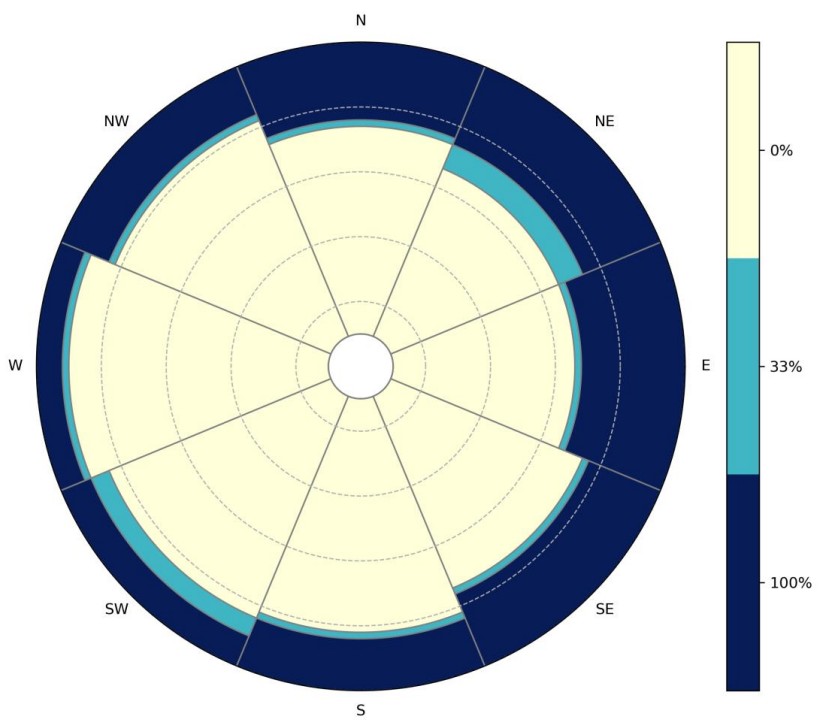


**Figure 7: Parametric Trigger for Mt. Fuji Each dashed line correspond to a unit of 10Km**





**Author contribution:**
Delioma Oramas-Dorta built the volcano risk model,  produced the risk results ("ELT") associated to the portfolio of
residential properties used in the Application, and researched and defined the physical trigger parameters for the
design of the volcano risk transfer mechanism presented in the paper. Giulio Tirabassi contributed to the definition
of the physical trigger parameters, and coded the mathematical design and optimization of the trigger.  Guillermo E.
Franco developed the original code as applied to earthquakes, and oversaw the adaptation of the code to the case of
volcanic eruptions. Christina Magill produced the tephra fall footprints used in the hazard module of the volcano
risk model, while working at Risk Frontiers.
**Acknowledgements:**
We would like thanking Guy Carpenter for permitting the use of its proprietary Volcano Risk Model for Six
Volcanoes in Japan, in order to produce the risk/ loss estimates this study used as a basis to design a parametric risk
transfer solution for volcanic eruptions. We would like to acknowledge the providers of several datasets that form
part of this Volcano Risk Model. In particular, Risk Frontiers (https://riskfrontiers.com/ ) provided the set of
stochastic volcanic tephra fall footprints that are part of the volcano risk model's hazard module. These footprints
were produced in 2017 following commission from Guy Carpenter, to form part of its proprietary Volcano Risk
Model for Six Volcanoes in Japan. Development of volcanic tephra fall footprints by Risk Frontiers used wind
reanalysis data (NCEP-DOE Reanalysis 2) from NOAA/OAR/ESRL PSD, Boulder, Colorado, USA
(https://www.esrl.noaa.gov/psd/). Rainfall data that also form part of the model's hazard module were provided by
JBA Risk Management, www.jbarisk.com.