# Peer review of "Design of parametric risk transfer solutions for volcanic"

_Natural Hazards and Earth System Sciences, 2019_

## Referee Comment (RC1) · Anonymous Referee #1 · 2 Jul 2019

The paper by Oramas-Dorta et al. is quite interesting because of its innovative use of the parametric risk transfer function as applied to the most widespread phenomenon associated with volcanic eruptions – tephra fallout. The paper is innovative in using a numerical model to inform the scale of index based payments, in this case essentially indexed by the VEI from one of six volcanoes in the metropolitan area of Tokyo.

I think the paper will be highly cited because of the great potential for using an index approach to hazard and risk assessments. The paper is quite clearly written, the mathematics clear, and the figures well presented. Although the paper deals with a specific scenario (ash impact on an urban area), it is the most likely scenario to have

widespread application. Therefore, I feel the paper is acceptable in its current form. I do have some suggestions for the authors, which I think can strengthen the manuscript further.

1. I am surprised that total eruption mass is not found to be a sensitive indicator of loss and does not appear in the parametric trigger design. Obviously, for explosive eruptions there is some correlation between eruption column height and loss, but not necessarily. For example, the Eyjall (Iceland) eruption mentioned in the intro did not have a particularly high plume, yet caused loss (although not for buildings – the focus of this paper). Does the point cloud shown in figure 3 collapse significantly (or is it significantly different) for eruption mass rather than plume height?

2. Similarly, eruption duration has a significant impact on loss and might be a useful part of the parametric trigger design. Unlike earthquakes, volcanic eruptions may have significant duration (years). The eruption duration not only impacts total load (and the ability to remove the load) but also the sectors (N,NE, etc.) likely to be impacted by the eruptions. Some mention of variable duration and its complicated influence on risk is warranted.

3. Plume height is measured remotely by satellite, and so fulfills a requirement of parametric trigger design to be quickly calculated and unbiased, compared with eruption mass. I think you should cite some important literature on this, like:

Prata, A.J. and Grant, I.F., 2001. Retrieval of microphysical and morphological properties of volcanic ash plumes from satellite data: Application to Mt Ruapehu, New Zealand. Quarterly Journal of the Royal Meteorological Society, 127(576), pp.2153-2179.

Pardini, F., Burton, M., Arzilli, F., La Spina, G. and Polacci, M., 2018. SO2 emissions, plume heights and magmatic processes inferred from satellite data: The 2015 Calbuco eruptions. Journal of Volcanology and Geothermal Research, 361, pp.12-24.

Merucci, L., Zakšek, K., Carboni, E. and Corradini, S., 2016. Stereoscopic estimation of volcanic ash cloud-top height from two geostationary satellites. Remote Sensing, 8(3), p.206.

4. One of the authors, C. Magill, has an important paper on tephra modeling in the Toyko region using Tephra2 to forecast loss. It is important to cite that paper because it provides essential groundwork for using Tephra2 to make these models, which is not covered in the current manuscript, whereas the current manuscript goes much farther in terms of illustrating a workflow for designing the parametric trigger.

Magill, C., Mannen, K., Connor, L., Bonadonna, C. and Connor, C., 2015. Simulating a multi-phase tephra fall event: inversion modelling for the 1707 Hoei eruption of Mount Fuji, Japan. Bulletin of Volcanology, 77(9), p.81.

5. In addition to VEI, you might mention alternative eruption scales, like magnitude. See:

Pyle, D.M., 2015. Sizes of volcanic eruptions. In The encyclopedia of volcanoes (pp. 257-264). Academic Press.

Rougier, J., Sparks, R.S.J., Cashman, K.V. and Brown, S.K., 2018. The global magnitude–frequency relationship for large explosive volcanic eruptions. Earth and Planetary Science Letters, 482, pp.621-629.

Just a few detailed comments:

Line 162. Change Kg to kg (lower case). Elsewhere in the paper, some units are capitalized. They should always be lower case. Line 163. Instead of saying vertical wind speed, say variation in wind speed with height in the atmosphere. Around line 293 – what is the relationship of eruption column height with total mass and eruption duration? Around line 510: it seems to me there is a fundamental difference between tephra fallout and these other phenomena (lava flows, pdcs, etc.). Tephra causes variable loading (depending on the eruption magnitude) so it seems more analogous

to earthquake damage. The other phenomena cause complete destruction to property in their path. So how does this influence the parametric trigger design? It must be binary for these other phenomena? Wrap this discussion back to the equations you present.
* * *

---

## Author Comment (AC1) · 22 Jan 2020

We thank Referee #1 for very useful questions and suggestions, to which we have replied below:

1. I am surprised that total eruption mass is not found to be a sensitive indicator of loss and does not appear in the parametric trigger design. Obviously, for explosive eruptions there is some correlation between eruption column height and loss, but not necessarily. For example, the Eyjall (Iceland) eruption mentioned in the intro did not have a particularly high plume, yet caused loss (although not for buildings – the focus of this paper). Does the point cloud shown in figure 3 collapse significantly (or is it

[Figure]

significantly different) for eruption mass rather than plume height?

Authors: We have produced a graph equivalent to that of Figure 3 of the manuscript (please see Fig.1 below), showing the relationship between total eruption mass and modeled loss. Comparison between this Figure (please find attached) and Figure 3 of the paper shows that eruption mass is, as rightly pointed out by the reviewer, a sensible indicator of loss. The reason that eruption mass does not appear in the parametric design, however, is because it does not fulfill the requisite of being obtainable on a near-real time basis (condition number 2 in Section 3) - even though it does fulfill conditions 1 and 3 mentioned in the Section. Whereas eruption column height is readily observable and can be objectively measured and reported on a real-time basis (as currently done by JMA), measurement/ estimation of eruption mass is not currently performed and reported on a real time basis. The parametric design, on the other hand and by definition, expects a non-perfect-relationship between the value of the chosen physical parameter and the resulting loss, which is incorporated in the basis risk (Section 3.2).

2. Similarly, eruption duration has a significant impact on loss and might be a useful part of the parametric trigger design. Unlike earthquakes, volcanic eruptions may have significant duration (years). The eruption duration not only impacts total load (and the ability to remove the load) but also the sectors (N,NE, etc.) likely to be impacted by the eruptions. Some mention of variable duration and its complicated influence on risk is warranted.

Authors: This is a very important observation and indeed the duration of the eruption should prove a significant driver of the loss. The reason why it wasn't included in the parametric design, however, is because it does not fulfill condition number 3 in Section 3 (eruption duration is not part of the stochastic event set in the catastrophe risk model developed). In this case, indeed, it is not possible to show the relationship between loss and eruption duration because the data is not available (contrary to the earlier case of loss versus eruption mass), although on the other hand a degree of correlation
between eruption duration and total eruption mass is expected. Future development of more complex and complete eruption catastrophe risk models should enable further investigation of alternative parametric designs for volcanic eruptions, using different – or a combination of different- triggers. We believe it is important however to discuss these issues in the current paper and will add comment in this respect.

3. Plume height is measured remotely by satellite, and so fulfills a requirement of parametric trigger design to be quickly calculated and unbiased, compared with eruption mass. I think you should cite some important literature on this, like:

Prata, A.J. and Grant, I.F., 2001. Retrieval of microphysical and morphological properties of volcanic ash plumes from satellite data: Application to Mt Ruapehu, New Zealand. Quarterly Journal of the Royal Meteorological Society, 127(576), pp.2153-2179.

Pardini, F., Burton, M., Arzilli, F., La Spina, G. and Polacci, M., 2018. SO2 emissions, plume heights and magmatic processes inferred from satellite data: The 2015 Calbuco eruptions. Journal of Volcanology and Geothermal Research, 361, pp.12-24. Merucci, L., Zakšek, K., Carboni, E. and Corradini, S., 2016. Stereoscopic estimation of volcanic ash cloud-top height from two geostationary satellites. Remote Sensing, 8(3), p.206.

Authors: Thank you very much for pointing this work out and will include.

4. One of the authors, C. Magill, has an important paper on tephra modeling in the Toyko region using Tephra2 to forecast loss. It is important to cite that paper because it provides essential groundwork for using Tephra2 to make these models, which is not covered in the current manuscript, whereas the current manuscript goes much farther in terms of illustrating a workflow for designing the parametric trigger. Magill, C., Mannen, K., Connor, L., Bonadonna, C. and Connor, C., 2015. Simulating a multiphase tephra fall event: inversion modelling for the 1707 Hoei eruption of Mount Fuji, Japan. Bulletin of Volcanology, 77(9), p.81.

Authors: Absolutely- it makes sense including this reference.

5. In addition to VEI, you might mention alternative eruption scales, like magnitude. See:

Pyle, D.M., 2015. Sizes of volcanic eruptions. In The encyclopedia of volcanoes (pp. 257-264). Academic Press. Rougier, J., Sparks, R.S.J., Cashman, K.V. and Brown, S.K., 2018. The global magnitude–frequency relationship for large explosive volcanic eruptions. Earth and Planetary Science Letters, 482, pp.621-629.

Authors: Thank you for pointing this out and will include in the eruption size discussion.

6. Just a few detailed comments:

Line 162. Change Kg to kg (lower case). Elsewhere in the paper, some units are capitalized. They should always be lower case.

Authors: thanks for pointing out.

Line 163. Instead of saying vertical wind speed, say variation in wind speed with height in the atmosphere.

Authors: the original sentence ("The model takes into account appropriate vertical wind speed and direction profiles") is not clear, we referred to "vertical profiles of both wind speed and direction". We can re-write as this, or else" variation in wind speed and direction with height in the atmosphere".

Around line 293 – what is the relationship of eruption column height with total mass and eruption duration?

Authors: We will include commentary on this relationship, as per discussion following from reviewer's comments 1 and 2. In this case we can look at the relationship between column height and eruption mass in a quantitative manner and from a qualitative stand-point in the case of the relationship with eruption duration since data on the latter is not available.

Around line 510: it seems to me there is a fundamental difference between tephra fallout and these other phenomena (lava flows, pdcs, etc.). Tephra causes variable loading (depending on the eruption magnitude) so it seems more analogous to earthquake damage. The other phenomena cause complete destruction to property in their path. So how does this influence the parametric trigger design? It must be binary for these other phenomena? Wrap this discussion back to the equations you present.

Authors: This is an interesting and thought provoking observation. Whereas tephra fallout can be considered as a gradually varying phenomenon that causes varying levels of damage, volcanic mass flows tend to produce either a total loss (assets in their path) or no loss (assets away from their path). The present work focuses solely on the design of a parametric trigger for tephra fallout, which has adopted the form of a Multilayer trigger in this particular study (Section 3.2). Regarding the potential design of a parametric trigger for volcanic mass flows, this is something that would have to be thoroughly investigated in future work. It may be the case that a Binary trigger (Section 3.2) would be appropriate; however, it is our view that a Multilayer trigger cannot be ruled out in principle, and that the binary nature of the damage/loss does not necessarily warrant the selection of a Binary trigger over a Multilayer trigger. It is our view that the design of a parametric trigger for these volcanic phenomena will substantially be determined by the characteristics of the physical modelling methodology applied.

Event losses for different predominant eruptive plume directions

Per- event loss (Bn. JPY)

Total eruption mass (Kg)

**Fig. 1.**

---

## Referee Comment (RC2) · Anonymous Referee #2 · 23 Sep 2020

In this paper, a novel method is developed to design a parametric risk transfer mechanism to offset losses from large, ash fall-producing volcanic eruptions. An application is shown for the case of Mount Fuji in Japan. The approach taken is novel and provides interesting advances scientifically and potentially for practice. The manuscript is very well written and the language of a high quality. My review focuses mainly on the methodological aspects and discussion of the results. In terms of volcanic eruptions per se, this is outside my own role of expertise. However, I discussed also this aspect with a volcano-specialist, whose opinion is that the main points related to that aspect have already been raised by reviewer 1. Overall, I believe that the paper could be accepted subject to several minor revisions. Specific review points can be found

below.

Introduction gives a clear introduction to the kinds of parametric insurance being covered in the paper, which is useful for the non-specialist reader. Also the description of the parametric insurance and selected triggers is very clear.

Some of the arguments in the introduction should be more clearly supported by evidence from the literature. For example, on line 100-102, provide literature to support the statement about the proper choice of parameters.

Wet version: on lines 165-171, the authors describe how they developed the "wet version" of the scenarios. They refer to a paper by Macedonio and Costa (2012) for the approach. Whilst this is fine, a short overview of this methods should also be summarized in this paper to give the reader an overall understanding of how it works (referring the reader to the paper for the details of course).

Vulnerability functions: Figure 2 gives a clear example of two vulnerability curves. However, for reproducibility, have the authors considered providing all curves, for example in a supplementary dataset?

BE module: please provide more information on how this is done – for example, how does the assignment on the probabilistic basis work?

Parts of the current conclusion would better split out into a separate discussion section. In particular, the parts discussing the limitations and challenges, as well as applicability elsewhere. This would give the opportunity to slightly expand these aspects, with reference to key literature. For example, given the topic of the special issue, one of two extra paragraphs describing key challenges for upscaling globally would be useful (there is some reasoning along this line but it is very short). The conclusion could then be kept shorter and more succinct.

---

## Author Comment (AC2) · 4 Nov 2020

We thank Referee #2 for their helpful feedback; please find our replies below:

1. Some of the arguments in the introduction should be more clearly supported by evidence from the literature. For example, on line 100-102, provide literature to support the statement about the proper choice of parameters.

Authors: Further background and references on this topic are provided below:

"Properly chosen parameters that are easy to measure transparently and with accuracy can provide parametric cat bonds with a speed of payment unparalleled in the domain

[Figure]

of insurance. The choice of parameters has evolved since the 1990's when these tools first appeared, resulting in different choices of design. For instance, in the case of earthquake two types of solutions have been used in the market successfully: first generation CAT-in-a-box triggers, and second-generation parametric indices (Franco 2010). The first type is based on the magnitude, epicenter location, and focal depth of the event, whereas the second are based on geographically distributed earthquake parameters such as ground motions. Second-generation are considered to be superior to first generation triggers owing to better correlation between the distributed parameters and resulting losses (Franco 2010, Goda 2013).

2. Wet version: on lines 165-171, the authors describe how they developed the "wet version" of the scenarios. They refer to a paper by Macedonio and Costa (2012) for the approach. Whilst this is fine, a short overview of this methods should also be summarized in this paper to give the reader an overall understanding of how it works (referring the reader to the paper for the details of course).

Authors: We propose to re-write the paragraph as follows:

"The methodology used to create "wet" footprints follows that described by Macedonio and Costa, 2012, whereby deposited ash fall increases its weight up to the point it becomes saturated with rainfall water, assuming a density of 1000 Kg/m3 and a total porosity of 60% for deposited ash fall from Mt. Fuji. Following Macedonio and Costa, 2012, we assume that all pores and interstices of the deposit are filled with water (water saturation), if enough water is available from a specific rainfall event. Rainfall data were supplied by JBA Risk Management in the form of 10,000 years of simulated daily precipitation that incorporates tropical cyclone and non-tropical cyclone precipitation."

3. Vulnerability functions: Figure 2 gives a clear example of two vulnerability curves. However, for reproducibility, have the authors considered providing all curves, for example in a supplementary dataset?

Authors: The source of the damage functions has been specified and referenced in the

paper (GAR15 Regional Vulnerability Functions report by Maqsood et al., 2015), which contains a comprehensive Annex with graphs for all the ash fall damage functions by construction type, building rise and roof pitch.

4. BE module: please provide more information on how this is done – for example, how does the assignment on the probabilistic basis work?

Authors: We propose to add the following text from line 219 onwards:

"To illustrate how the BE works, let us take an example of a Residential building in a Postal Code in Kanagawa prefecture. If that is all the information we know about this asset, the BE module will use the weights corresponding to Residential buildings in that postal code to assign a specific location within the postal code and a set of characteristics (construction type, etc.) to this Residential building (please see Table 2 for a list of possible Residential building types). Such assignation is probabilistic in the sense that a distribution of likely locations and characteristics will be generated for each risk, through iterative sampling based on those weights. Such distribution will eventually be propagated to the loss calculation part of the model, in order to produce a final loss distribution for this building."

5. Parts of the current conclusion would better split out into a separate discussion section. In particular, the parts discussing the limitations and challenges, as well as applicability elsewhere. This would give the opportunity to slightly expand these aspects, with reference to key literature. For example, given the topic of the special issue, one of two extra paragraphs describing key challenges for upscaling globally would be useful (there is some reasoning along this line but it is very short). The conclusion could then be kept shorter and more succinct.

Authors: We propose to re-write as follows (additional reference have been added at the end):

[revised manuscript text omitted]

---

## Author Response (AR1)

**1 Author's Response**

**2 Reviewer #1:**

1. I am surprised that total eruption mass is not found to be a sensitive indicator of loss and does not appear in
the parametric trigger design. Obviously, for explosive eruptions there is some correlation between eruption
column height and loss, but not necessarily. For example, the Eyjall (Iceland) eruption mentioned in the intro did
not have a particularly high plume, yet caused loss (although not for buildings – the focus of this paper). Does
the point cloud shown in figure 3 collapse significantly (or is it significantly different) for eruption mass rather than
plume height?

**Authors**: We have produced a graph equivalent to that of Figure 3 of the manuscript, showing the relationship
between total eruption mass and modelled loss (please see below). Comparison between this Figure and Figure
3 shows that eruption mass is, as rightly pointed out by the reviewer, a sensible indicator of loss. The reason that
eruption mass does not appear in the parametric design, however, is because it does not fulfill the requisite of
being obtainable on a near-real time basis (condition number 2 in Section 3) - even though it does fulfill conditions
1 and 3 mentioned in the Section. Whereas eruption column height is readily observable and can be objectively
measured and reported on a real-time basis (as currently done by JMA), measurement/ estimation of eruption
mass is not currently performed and reported on a real time basis. The parametric design, on the other hand and
by definition, expects a non-perfect-relationship between the value of the chosen physical parameter and the
resulting loss, which is incorporated in the basis risk (Section 3.2).

[Figure]

2. Similarly, eruption duration has a significant impact on loss and might be a useful part of the parametric trigger
design. Unlike earthquakes, volcanic eruptions may have significant duration (years). The eruption duration not
only impacts total load (and the ability to remove the load) but also the sectors (N,NE, etc.) likely to be impacted
by the eruptions. Some mention of variable duration and its complicated influence on risk is warranted.

**Authors**: This is a very important observation and indeed the duration of the eruption should prove a significant driver of the loss. The reason why it wasn't included in the parametric design, however, is because it does not fulfill condition number 3 in Section 3 (eruption duration is not part of the stochastic event set in the catastrophe risk model developed). In this case, indeed, it is not possible to show the relationship between loss and eruption duration because the data is not available (contrary to the earlier case of loss versus eruption mass), although on the other hand a degree of correlation between eruption duration and total eruption mass is expected. Future development of more complex and complete eruption catastrophe risk models should enable further investigation of alternative parametric designs for volcanic eruptions, using different –or a combination of different- triggers. We believe it is important however to discuss these issues in the current paper and have added comment in this respect (line 312+ of pdf Manuscript).

3. Plume height is measured remotely by satellite, and so fulfills a requirement of parametric trigger design to be quickly calculated and unbiased, compared with eruption mass. I think you should cite some important literature on this, like:

Prata, A.J. and Grant, I.F., 2001. Retrieval of microphysical and morphological properties of volcanic ash plumes from satellite data: Application to Mt Ruapehu, New Zealand. Quarterly Journal of the Royal Meteorological Society, 127(576), pp.2153- 2179.

Pardini, F., Burton, M., Arzilli, F., La Spina, G. and Polacci, M., 2018. SO2 emissions, plume heights and magmatic processes inferred from satellite data: The 2015 Calbuco eruptions. Journal of Volcanology and Geothermal Research, 361, pp.12-24. Merucci, L., Zakšek, K., Carboni, E. and Corradini, S., 2016. Stereoscopic estimation of volcanic ash cloud-top height from two geostationary satellites. Remote Sensing, 8(3), p.206.

**Authors**: Thank you very much for pointing this work out and have included (line 535 of pdf Manuscript).

4. One of the authors, C. Magill, has an important paper on tephra modeling in the Toyko region using Tephra2 to forecast loss. It is important to cite that paper because it provides essential groundwork for using Tephra2 to make these models, which is not covered in the current manuscript, whereas the current manuscript goes much farther in terms of illustrating a workflow for designing the parametric trigger. Magill, C., Mannen, K., Connor, L., Bonadonna, C. and Connor, C., 2015. Simulating a multi-phase tephra fall event: inversion modelling for the 1707 Hoei eruption of Mount Fuji, Japan. Bulletin of Volcanology, 77(9), p.81.

**Authors**: Absolutely- it makes sense including this reference (line 169 of pdf Manuscript).

5. In addition to VEI, you might mention alternative eruption scales, like magnitude. See:

Pyle, D.M., 2015. Sizes of volcanic eruptions. In The encyclopedia of volcanoes (pp. 257-264). Academic Press. Rougier, J., Sparks, R.S.J., Cashman, K.V. and Brown, S.K., 2018. The global magnitude–frequency relationship for large explosive volcanic eruptions. Earth and Planetary Science Letters, 482, pp.621-629.

**Authors**: Thank you for pointing this out and have included (footnote #1 of pdf Manuscript)

6. Just a few detailed comments:

- Change Kg to kg (lower case). Elsewhere in the paper, some units are capitalized. They should always be lower case.

   **Authors**: thanks for pointing out, changes made throughout text

- Instead of saying vertical wind speed, say variation in wind speed with height in the atmosphere.

**Authors**: the original sentence ("The model takes into account appropriate vertical wind speed and
direction profiles") is not clear, we referred to "vertical profiles of both wind speed and direction". We can
have re-written in line 170 of pdf Manuscript.
• Around line 293 – what is the relationship of eruption column height with total mass and eruption
duration?
**Authors**: We have included commentary on this relationship, as per discussion following from reviewer's
comments 1 and 2 (line 312+ of pdf Manuscript).
• Around line 510: it seems to me there is a fundamental difference between tephra fallout and these other
phenomena (lava flows, pdcs, etc.). Tephra causes variable loading (depending on the eruption
magnitude) so it seems more analogous to earthquake damage. The other phenomena cause complete
destruction to property in their path. So how does this influence the parametric trigger design? It must
be binary for these other phenomena? Wrap this discussion back to the equations you present.
**Authors**: This is an interesting and thought provoking observation. Whereas tephra fallout can be
considered as a gradually varying phenomenon that causes varying levels of damage, volcanic mass
flows tend to produce either a total loss (assets in their path) or no loss (assets away from their path).
The present work focuses solely on the design of a parametric trigger for tephra fallout, which has
adopted the form of a Multilayer trigger in this particular study (Section 3.2). Regarding the potential
design of a parametric trigger for volcanic mass flows, this is something that would have to be
thoroughly investigated in future work. It may be the case that a Binary trigger (Section 3.2) would be
appropriate; however, it is our view that a Multilayer trigger cannot be ruled out in principle, and that
the binary nature of the damage/loss does not necessarily warrant the selection of a Binary trigger
over a Multilayer trigger. It is our view that the design of a parametric trigger for these volcanic
phenomena will substantially be determined by the characteristics of the physical modelling
methodology applied.

## Reviewer #2:

1. Some of the arguments in the introduction should be more clearly supported by evidence from the literature.
For example, on line 100-102, provide literature to support the statement about the proper choice of parameters.
**Authors**: Further background and references on this topic have been provided in line 100+ of pdf Manuscript.
2. Wet version: on lines 165-171, the authors describe how they developed the "wet version" of the scenarios.
They refer to a paper by Macedonio and Costa (2012) for the approach. Whilst this is fine, a short overview of
this methods should also be summarized in this paper to give the reader an overall understanding of how it works
(referring the reader to the paper for the details of course).
**Authors**: Further details have been provided in line 176+ of pdf Manuscript.
3. Vulnerability functions: Figure 2 gives a clear example of two vulnerability curves. However, for reproducibility,
have the authors considered providing all curves, for example in a supplementary dataset?
**Authors**: The source of the damage functions has been specified and referenced in the paper (GAR15 Regional
Vulnerability Functions report by Maqsood et al., 2015), which contains a comprehensive Annex with graphs for
all the ash fall damage functions by construction type, building rise and roof pitch.
4. BE module: please provide more information on how this is done – for example, how does the assignment on
the probabilistic basis work?
**Authors**: Further details have been provided in line 227+ of pdf Manuscript.

5. Parts of the current conclusion would better split out into a separate discussion section. In particular, the parts discussing the limitations and challenges, as well as applicability elsewhere. This would give the opportunity to slightly expand these aspects, with reference to key literature. For example, given the topic of the special issue, one of two extra paragraphs describing key challenges for upscaling globally would be useful (there is some reasoning along this line but it is very short). The conclusion could then be kept shorter and more succinct.

**Authors**: The Conclusions section has been split into Discussion and Conclusions as advised (line 503+ of pdf Manuscript), and these topics have been expanded with the following additional references added:

[revised manuscript text omitted]
 | $9.84\times10^{-8}$ | $1.03\times10^{12}$ | $1.28\times10^{9}$ | $1.32\times10^{11}$ |
| 1589 | Fuji | $3.65\ \times10^{-7}$ | $1.87\times10^{6}$ | $2.25\times10^{6}$ | $1.93\times10^{7}$ |
| 1590 | Fuji | $4.91\times10^{-8}$ | $1.36\times10^{13}$ | $4.29\times10^{9}$ | $1.01\times10^{12}$ |
| 1591 | Fuji | $9.82\times10^{-7}$ | 0 | 0 | 0 |

**Table 5: Subset of ELT outputs from the volcano risk model, run of the residential portfolio described. The table shows losses on the portfolio caused by four of the model's ash fall events from Mt. Fuji. The mean loss and the standard deviation of the loss distribution associated to each event (in JPY) are reported in the ELT.**

| | Plume Height Thresholds [kKm] | | | | | | | | Yearly Exceedance Probability | Transferred Risk | Layer Payment |
|---|---|---|---|---|---|---|---|---|---|---|---|
| | N | NE | E | SE | S | SW | W | NW | | | |
| Layer 1 | 32 | 28 | 28 | 32 | 36 | 37 | 40 | 36 | 0.026% | 76% | 33% |
| Layer 2 | 33 | 32 | 29 | 33 | 37 | 40 | 41 | 37 | 0.020% | 67% | 100% |

**Table 6: Parametric trigger for Mt Fuji. The risk transferred by each layer is expressed as percentage over the total risk of**
**Mt Fuji. The layer payment is expressed as fraction of the maximum payment (300 Billion JPY).**

**Figures**

[Figure]

**Figure 1: The geographic domain of the volcano ash fall model presented in this paper includes Tokyo and Kanagawa**
**Prefectures in Japan, and the six major volcanoes that can affect them, Fuji, Hakone, Asama, Haruna, Kita-Yatsugatake,**
**and Kusatsu-Shirane.**

[Figure]

**Figure 2: Damage functions for two different building types considered in the volcano risk model ("RC-SRC" stands for**
**Reinforced Concrete- Steel Reinforced Concrete; "Med." stands for Medium); source of these damage functions is Maqsood**
**et al., 2014.**

[Figure]

**Figure 3: Relationship between height of eruptive column (in Km, from crater rim) and modelled losses for all eruptive events in the volcano risk model. Each panel displays a subset of eruptions featuring a specific predominant direction of their eruptive plume (East, North, North-East, North-West, South, South-East, South-West and West).**

[Figure]

**Figure 4: Pareto front for a binary trigger designed modelling stochastic losses for Mt. Fuji. The transferred risk is**
**displayed as percentage of the total risk.**

[Figure]

**Figure 5: (Left) Modelled AAL for the six volcanoes included in the volcano risk model. (Right) Breakdown of Mt Fuji risk**
**by wind sector.**

[Figure]

**Figure 6: OEP curve for Mt Fuji losses (blue) and trigger payments (orange)**

[Figure]

**Figure 7: Parametric Trigger for Mt. Fuji Each dashed line correspond to a unit of 10Km**

**Author contribution:**

Delioma Oramas-Dorta built the volcano risk model, produced the risk results ("ELT") associated to the portfolio of
residential properties used in the Application, and researched and defined the physical trigger parameters for the design
of the volcano risk transfer mechanism presented in the paper. Giulio Tirabassi contributed to the definition of the
physical trigger parameters, and coded the mathematical design and optimization of the trigger. Guillermo E. Franco
developed the original code as applied to earthquakes, and oversaw the adaptation of the code to the case of volcanic
eruptions. Christina Magill produced the tephra fall footprints used in the hazard module of the volcano risk model,
while working at Risk Frontiers.

**Acknowledgements:**

We would like thanking Guy Carpenter for permitting the use of its proprietary Volcano Risk Model for Six Volcanoes
in Japan, in order to produce the risk/ loss estimates this study used as a basis to design a parametric risk transfer
solution for volcanic eruptions. We would like to acknowledge the providers of several datasets that form part of this
Volcano Risk Model. In particular, Risk Frontiers (https://riskfrontiers.com/ ) provided the set of stochastic volcanic
tephra fall footprints that are part of the volcano risk model's hazard module. These footprints were produced in 2017
following commission from Guy Carpenter, to form part of its proprietary Volcano Risk Model for Six Volcanoes in
Japan. Development of volcanic tephra fall footprints by Risk Frontiers used wind reanalysis data (NCEP-DOE
Reanalysis 2) from NOAA/OAR/ESRL PSD, Boulder, Colorado, USA *(https://www.esrl.noaa.gov/psd/). Rainfall data
that also form part of the model's hazard module were provided by JBA Risk Management, www.jbarisk.com.